# Silencing *miR-146a-5p* Protects against Injury-Induced Osteoarthritis in Mice

**DOI:** 10.3390/biom13010123

**Published:** 2023-01-07

**Authors:** Haocheng Qin, Cuicui Wang, Yonghua He, Aiwu Lu, Tiandao Li, Bo Zhang, Jie Shen

**Affiliations:** 1Department of Orthopaedic Surgery, School of Medicine, Washington University, St. Louis, MO 63110, USA; 2The 2nd Xiangya Hospital, Central South University, Changsha 410021, China; 3Department of Developmental Biology, Center of Regenerative Medicine, Washington University, St. Louis, MO 63110, USA

**Keywords:** osteoarthritis, articular chondrocyte, inflammation, *miR-146a-5p*

## Abstract

Osteoarthritis (OA), the most prevalent joint disease and the leading cause of disability, remains an incurable disease largely because the etiology and pathogenesis underlying this degenerative process are poorly understood. Low-grade inflammation within joints is a well-established factor that disturbs joint homeostasis and leads to an imbalance between anabolic and catabolic processes in articular cartilage; however, the complexity of the network between inflammatory factors that often involves positive and negative feedback loops makes current anti-cytokine therapy ineffective. MicroRNAs (miRNAs) have emerged as key regulators to control inflammation, and aberrant miRNAs expression has recently been linked to OA pathophysiology. In the present study, we characterized transcriptomic profiles of miRNAs in primary murine articular chondrocytes in response to a proinflammatory cytokine, IL-1β, and identified *miR-146a-5p* as the most responsive miRNA to IL-1β. *miR-146a-5p* was also found to be upregulated in human OA cartilage. We further demonstrated that knockdown of *miR-146a-5p* antagonized IL-1β-mediated inflammatory responses and IL-1β-induced catabolism in vitro, and silencing of *miR-146a* in chondrocytes ameliorated articular cartilage destruction and reduced OA-evoked pain in an injury-induced murine OA model. Moreover, parallel RNA sequencing revealed that differentially expressed genes in response to IL-1β were enriched in pathways related to inflammatory processes, cartilage matrix homeostasis, and cell metabolism. Bioinformatic analyses of putative *miR-146a-5p* gene targets and following prediction of protein–protein interactions suggest a functional role of *miR-146a-5p* in mediating inflammatory processes and regulation of cartilage homeostasis. Our genetic and transcriptomic data define a crucial role of *miR-146a-5p* in OA pathogenesis and implicate modulation of *miR-146a-5p* in articular chondrocytes as a potential therapeutic strategy to alleviate OA.

## 1. Introduction

Osteoarthritis (OA) is the most prevalent musculoskeletal disease in the Medicare population and has reached epidemic proportions not only in the United States but also worldwide [1,2]. Despite many well-characterized risk factors, effective disease-modifying therapeutics for treatment of OA are currently not available due to limited understanding of the disease cause and pathogenesis; as such, pain relief medications and surgical joint replacement remain the major treatments for advanced OA [1,3].

Although OA is currently viewed as a whole-joint disorder, articular cartilage degeneration remains the primary pathological change in altering joint function. The imbalance between anabolic and catabolic responses at cellular and molecular levels contributes to the etiology of OA, shifting normal cartilage homeostasis towards a catabolic state [4]. A number of factors have been implicated to participate in this process. Several growth factors that favor anabolic actions, such as TGFβ and IGF-1, have been identified as essential regulators for maintaining cartilage function and integrity [5,6,7]. In contrast, proinflammatory cytokines, particularly IL-1β and TNF-α, are detrimental and trigger a series of catabolic activities, leading to loss of tissue homeostasis and degradation of articular cartilage [8,9]. Due to their abilities to induce a variety of signaling pathways and significant proinflammatory and catabolic effects in promoting the progression of OA, suppression of these cytokines have been studied as a potential therapeutic treatment of OA and to prevent progressive cartilage loss. However, the desired effects of stopping OA progression are not achieved by simply blocking a single proinflammatory cytokine [10,11,12]. Considering the common signaling pathways activated by different cytokines, the negative results obtained from anti-cytokine therapy led to the notion that identification of the effector molecules in inflamed cartilage may be crucial for discovering effective therapeutics and improving current treatments for OA.

Recently, microRNAs (miRNAs), a family of non-coding small RNAs, which regulate gene expression through targeting mRNAs for degradation or translational repression [13], have received increasing attention in regulating cartilage function and homeostasis [14,15,16,17]. It has become apparent that aberrant miRNA expression is associated with the OA pathologies, and the breadth of knowledge is continuously expanding. Indeed, using a literature search and study selection with certain criteria, Stanciugelu et al. showed that, in the last decade, a total of 54 differentially expressed miRNAs have been identified in human OA chondrocytes and/or cartilage [18]. Additionally, studies from animal models have demonstrated several miRNAs to be regulated by certain risk factors of OA, including excess mechanical stress, joint injury, and inflammation [19,20,21,22,23,24]. On the other hand, miRNAs have been highlighted in the post-transcriptional regulation of inflammatory mediators in a variety of disease processes, including OA [15,17,18]. Despite the correlative evidence linking dysregulated mRNA expression to OA, the role of miRNAs in the pathogenesis of OA remains yet to be elucidated. In the present study, we sought to examine how miRNAs are regulated in response to inflammation in articular chondrocytes and define their role in articular cartilage homeostasis.

Here, we used high-throughput microRNA sequencing (miRNA-Seq) to profile miRNA expression in primary murine articular chondrocytes in response to IL-1β. Of the 534 miRNAs profiled, the miRNA showing the greatest fold change was *miR-146a-5p*, which was upregulated by approximately four-fold by IL-1β. In vitro and ex vivo studies revealed that knockdown of *miR-146a-5p* suppressed IL-1β-mediated NF-κB activation and alleviated the imbalance between anabolism and catabolism caused by inflammation. We further investigated the cartilage-specific role of *miR-146a* in the pathophysiology of OA using genetic approaches combined with an injury-induced OA model in mice. Our in vivo data suggest a protective effect of *miR-146a* inhibition against OA, as evidenced by reduced inflammation, less cartilage degradation, and preserved tissue integrity. In addition, through an integrated analysis by combining gene expression profiling from high-throughput RNA sequencing (RNA-Seq) with a bioinformatics approach, our data highlighted the functional association of *miR-146a-5p* with genes and pathways involved in inflammation and cartilage matrix homeostasis. Collectively, our work provides comprehensive transcriptome-wise analyses of regulatory *miR-146a-5p* and its cartilage-specific regulation of mRNAs, and it implicates modulation of *miR-146a-5p* as a potential therapeutic strategy for OA.

## 2. Material and Methods

### 2.1. Mice and Human Tissues

All animal studies were performed in accordance with approval of the Committees on Animal Resources in Washington University in St. Louis. *Agc1Cre^ERT2^* mice [25] (JAX, #019148) and *miR-146a* floxed mice [26] (miR-146a^f/f^, JAX, #034342) were purchased from Jackson Laboratory (Jackson Laboratory, Bar Harbor, ME, USA). *miR-146a^f/f^* and *Agc1Cre^ERT2^;miR-146a^f/f^* mice were viable and produced in Mendelian ratios. Meniscal ligament injury (MLI) surgery was introduced unilaterally in the right knee joints of *miR-146a^f/f^* and *Agc1Cre^ERT2^;miR-146a^f/f^* male mice at 3 months of age as previously documented [27]. At 4 weeks post-surgery, tamoxifen was administered via intraperitoneal injection daily at a dose of 1 mg/10 g body weight for 5 consecutive days to examine any disease-modifying effect of *miR-146a* deletion on injury-induced OA progression.

Human cartilage tissue collection was approved by the Washington University Human Research Protection Office (IRB #201104119). Healthy articular cartilage tissues were provided by National Disease Research Interchange. OA articular cartilage tissues were obtained from patients undergoing total knee arthroplasty at Barnes-Jewish Hospital.

### 2.2. Histological Analyses

Mouse knees were collected at indicated time points and subjected to fixation, decalcification, processing, and sectioning. Knee sections were stained with safranin-O/fast green to analyze phenotypical changes in joints. Histological scoring of OA-like changes in the medial femoral condyle and tibial plateau of the knee joints were performed using the established Osteoarthritis Research Society International (OARSI) scoring system (score, 0–6) [28]. Immunohistochemistry staining for Col2a1 (1:100, Thermo Fisher Scientific, Waltham, MA, USA, #MS235-P), phospho-p65 (1:100, Cell Signaling, Danvers, MA, USA, #3033), Mmp13 (1:200, Abcam, Cambridge, UK, #ab39012), and Prg4 (1:200, Abcam, #ab28484) was conducted on paraffin sections following appropriate antigen retrieval methods. Signals were developed with DAB reagents (Vector Laboratories, Burlingame, CA, USA, #SK-4100) and counterstained with methyl green. Three sections of each specimen were examined for all histological analyses. Fluorescent in situ hybridization (FISH) specific for *miR-146a* was performed on paraffin sections of human cartilage specimens. In brief, the cartilage specimens were fixed by 4% paraformaldehyde for 20 min, followed with 15 min Proteinase K (10 μg/mL) incubation. Biotin-labeled probe (5′-TGA GAA CTG AAT TCC ATG GGT T-3′) was used for overnight incubation. After strevidine-HRP incubation, Tyramide 594 kit (Thermo Fisher Scientific, Waltham, MA, USA, #B40944) was used to develop fluorescence.

### 2.3. Micro-CT Analyses

Micro-CT analyses were performed on mouse knee joints prior to decalcification using a VivaCT 40 scanner. Briefly, the tibia was scanned from the knee to the connection at the fibula using a protocol consisting of high-resolution X-Ray energy settings of 55 kV and 145 μA and integration time of 300 ms. Parameters of subchondral bone, including tissue volume and bone volume, were measured using Scanco analysis software. For the knee joints that experienced MLI injuries, analyses were focused on the medial tibial plateau in order to reveal more apparent pathological changes.

### 2.4. Pain Studies

Electronic Von Frey (Bioseb, Pinellas Park, FL, USA, #Bio-EVF4) was used to determine the mechanical sensitivity threshold in mice post-MLI surgeries. The central area of murine hind paws was pressed against the electronic probe to induce nociceptive pain, and the intensity of the stimulus was automatically recorded when the paws were withdrawn. OA-evoked pain was further assessed using a Small Animal Algometer (SMALGO; Bioseb, Pinellas, FL, USA, #Bio-SMALGO). Progressive force was applied on the electronic probe against both uninjured and injured knees, and the force was automatically documented until the limbs were withdrawn. Maximum force was limited to 450 g to avoid damage of knee joints. For both Von Frey and SMALGO assays, each mouse was measured three independent times and then averaged as a readout.

### 2.5. Primary Articular Chondrocyte and Articular Cartilage Explant Cultures

Human articular chondrocytes were isolated from healthy donors and total knee replacement patients. Briefly, articular cartilages were dissected into small pieces and digested for 90 min by 0.4% pronase (Roche, Indianapolis, IN, USA, #10165921001) and 14 h by 0.035% collagenase P (Roche, #11249002001). Following filtering, human articular chondrocytes were seeded in high-glucose DMEM (Thermo Fisher Scientific, #10569010) with 10% FBS (Gibco, New York, NY, USA, #10437) at the density of 10^5^ cells/cm^2^.

Murine articular chondrocytes were isolated as previously described [29] with modifications. Briefly, articular cartilages were dissected from femoral heads of 3-week-old wild-type mice and digested for 4–6 h in 0.5 mg/mL collagenase P (Roche, Indianapolis, IN, USA. # 11249002001) in high-glucose DMEM (Thermo Fisher Scientific, Waltham, MA, USA, #10569010) supplemented with 1% penicillin/streptomycin. Following digestion, cells were collected and seeded at a density of 50 × 10^4^ cells/well or 25 × 10^4^ cells/well in 12- or 24-well plates, respectively, according to experimental designs. The next day, cells were treated with IL-1β (1 ng/mL) (R&D Systems, Minneapolis, MN, USA, #201-LB-005) for 24 h. To examine the knockdown efficacy of *miR-146a-5p*, cells were transfected with *miR-146a-5p* inhibitor (Thermo Fisher Scientific, Waltham, MA, USA, #4464084) at different concentrations in reduced serum media (Opti-MEM, Thermo Fisher Scientific, Waltham, MA, USA, #31985070) overnight. Cells were then cultured in complete media (high-glucose DMEM supplemented with 2% FBS [Gibco, #10437] and 1% penicillin/streptomycin), allowing cell recovery and gene expression. In experiments associated with suppression of *miR-146a-5p*, articular chondrocytes were pretreated with *miR-146a-5p* inhibitor overnight at a concentration as determined. IL-1β was then added to cells for another 24 h before proceeding to RNA extraction. For explant cultures, articular cartilages were similarly dissected from femoral heads and immediately transfected with *miR-146a-5p* inhibitor at same concentration used in cell cultures in reduced serum media overnight. Following transfection, explants were treated with IL-1β for 5 days, with media replaced every 2 days. After completion of the treatment course, explants were subjected to histological assessments.

### 2.6. Real-Time qPCR

RNA was extracted from human or murine articular chondrocytes using an RNeasy Mini kit (Qiagen, Hilden, Germany, #74134). cDNA synthesis (Bio-Rad, Hercules, CA, USA, #1708841) and real-time qPCR (Bio-Rad, #1725274) were performed according to the manufacturer’s instructions. Primer sequences are presented in Appendix A.

To examine the *miR-146a-5p* expression level, TaqMan miRNA reverse transcription kit (Thermo Fisher Scientific, Waltham, MA, USA, #4366596) and TaqMan master mix (Thermo Fisher Scientific, #4440040) were used following the manufacturer’s instructions. *miR-146a-5p* expression (Thermo Fisher Scientific, Waltham, MA, USA, assay ID #000468) was analyzed and normalized to U6 snRNA (Thermo Fisher Scientific, Waltham, MA, USA, assay ID #001973) in murine articular chondrocytes.

### 2.7. MicroRNA-Seq and RNA-Seq Assays and Data Analyses

RNA-Seq and MicroRNA-Seq (miRNA-Seq) were performed using primary murine articular chondrocytes treated with vehicle and IL-1β. Raw reads were processed in the same fashion, using an in-house pipeline and open-source R packages. Briefly, raw reads were first trimmed using cutadapt to remove low-quality bases and reads. Trimmed reads were then aligned to the mouse genome mm10 with GENCODE annotation vM20 using STAR (v2.5.4) with default parameters. Transcript quantification was performed using featureCounts from the subread package (v1.6.3). Further quality control assessments were made using RSeQC and RSEM, and batch correction was performed using RUVseq [30].

Principle component analysis (PCA) and differential expression analysis for vehicle and IL-1β-treated groups were determined using DESeq2 in negative binomial mode using batch-corrected transcripts from featureCounts (>2-fold expression change, >1 count per million (CPM), Benjamini corrected *p* < 0.05). Pairwise comparisons were made between IL-1β and vehicle to determine differentially expressed genes (DEGs). The gene expression was plotted using ggplot2 and pheatmap libraries in R environment.

An entire list of computationally predicted gene targets for *miR-146a-5p* was obtained from the TargetScan [31] web resource by searching for the presence of conserved 8mer, 7mer, and 6mer sites that match the seed region of *miR-146a-5p*. Putative targets of *miR-146a-5p* in response to IL-1β in articular chondrocytes were selected by overlaying the entire list of predicted targets with the 420 downregulated DEGs identified in RNA-Seq.

### 2.8. Pathway and Functional Association Analyses

DEGs from RNA-Seq with Log_2_FC > 2 and *Padj* < 0.05 were uploaded into the Enrichr [32] and overlaid with the global signaling pathways from KEGG database to identify significant pathways and related genes. The *miR-146a-5p* putative targets and the gene subset enriched in the most significant pathways were used to construct a protein–protein interaction network with STRING database [33]. Interaction sources from biochemical/genetic experiments and databases are only selected as the association evidence. Minimum required interaction score was set to 0.7 to identify potential protein–protein interactions with high confidence. Disconnected nodes in the network were hidden in the predicted functional association network.

### 2.9. Statistical Analyses

All data were expressed as means ± SD. Results were analyzed using GraphPad Prism version 7 (GraphPad Software, Inc., San Diego, CA, USA). Comparisons between groups were analyzed using two-tailed unpaired Student’s *t*-test. When comparing multiple groups, two-way ANOVA was used, followed by Tukey test as appropriate, for subsequent pairwise (group) comparisons. A *p* value < 0.05 was considered statistically significant.

## 3. Results

### 3.1. Inflammation Occurs in OA Cartilage and Exogenous Administration of Proinflammatory Cytokines Provokes Catabolic Changes in Articular Chondrocytes

To establish the link between inflammation and OA occurrence, we first assessed the levels of phospho-p65, an essential indicator of NF-κB signaling activation, in normal and OA cartilage from patients. Compared to the nearly undetected levels in normal cartilage, phospho-p65 was remarkably elevated in OA cartilage (Figure 1a). Interestingly, phospho-p65 was specifically seen in cell clusters residing in the superficial zone, a hallmark of arthritic cartilage, which express a broad range of differentiation markers and aberrant signaling activation related to disease pathogenesis [34]. In line with NF-κB activation in arthritic cartilage, inflammatory cytokines, including *IL6* and *TNFa*, were significantly upregulated in human OA articular chondrocytes compared to healthy cells (Figure 1b). In addition, one of the key cartilage-degrading enzymes, *MMP13*, was substantially increased in OA chondrocytes (Figure 1b), consistent with a shift towards a catabolic state occurring in progressive OA. These findings buttress the phenomenon that inflammation is one of the typical manifestations of OA.

Next, primary murine articular chondrocytes were isolated and treated with IL-1β, an activator of NF-κB and a proinflammatory cytokine closely associated with OA [35], and inflammatory factors and pathological proteases were evaluated. Compared to control, IL-1β markedly stimulated expression of cytokines, including *Il6* and *Tnfα* (Figure 1c). Moreover, *Mmp13* was greatly upregulated in the presence of IL-1β (Figure 1c). These results collectively suggest that proinflammatory mediators elicit inflammatory responses in articular chondrocytes, leading to excess production of proteolytic enzymes, accounting for cartilage degradation.

### 3.2. miR-146a-5p Is Induced in Response to IL-1β in Murine Articular Chondrocytes and Increased in Human OA Cartilage

To identify the miRNAs involved in OA pathogenesis, high-throughput miRNA-Seq was conducted to screen for miRNAs responsive to IL-1β treatment in primary murine articular chondrocytes. Expression profiling of 534 miRNAs identified 23 miRNAs that showed differential expression (Log_2_FC > 0.3, *Padj* < 0.05) between IL-1β- and vehicle-treated cells (Figure 2a, Appendix A). Among the most noticeable changes was *miR-146a-5p*, which was upregulated by nearly 4-fold (Figure 2a). This was further confirmed by real time qPCR for *miR-146-5p*, showing a similar degree of elevation in IL-1β-treated cells (Figure 2b). More importantly, using fluorescent in situ hybridization (FISH) specific for *miR-146a-5p*, a marked increase in *miR-146a-5p* was visualized in human OA cartilage, largely in chondrocyte clusters when overlapping with DAPI staining (Figure 2c). This miRNA is of particular interest given previous findings showing its abnormal elevation in OA chondrocytes and/or cartilage [22,36] and its association with susceptibility to a variety of inflammatory disorders [37]. For these reasons, *miR-146a-5p* was selected for further investigation of its biological role in OA development.

### 3.3. Knockdown of miRNA-146a-5p Antagonizes IL-1β-Induced Cytokines and Catabolic Makers of Articular Chondrocytes

To elucidate the functional role of *miR-146a-5p* in the pathogenesis of OA, we transfected primary murine articular chondrocytes with an inhibitor specific for *miR-146a-5p*. Expression of *miR-146a-5p* revealed a dose-dependent reduction at 25 and 50 nM of the *miR-146a-5p* inhibitor. Compared with control inhibitor, *miR-146a-5p* was decreased nearly 80% by transfection of 50 nM of the *miR-146a-5p* inhibitor without affecting cell viability. Nonetheless, higher concentration of the inhibitor did not result in further significant reduction in *miR-146a-5p* expression (Appendix A). Thus, *miR-146a-5p* inhibitor was set at 50 nM for the following studies.

The effect of miRNAs modulation on inflammatory and catabolic genes was next explored both at a basal level and following IL-1β induction. Basal *Il6* mRNA was greatly reduced (by 81%) in the presence of the *miR-146a-5p* inhibitor (Figure 3a). By contrast, abolishing of *miR-146a-5p* appeared to have no apparent effect on basal *Mmp13* mRNA levels (Figure 3b). Knockdown of *miR-146a-5p* significantly attenuated IL-1β inducted *Il6* by 44% (Figure 3a). Similarly, the elevation of *Mmp13* by IL-1β was antagonized by inhibition of *miR-146a-5p* (Figure 3b). The findings that suppression of *miR-146a-5p* reduced the IL-1β-induced *Il6* and *Mmp13* suggest that *miR-146a-5p* may play a detrimental role in OA pathogenesis.

The changes in gene expression elicited by *miR-146a-5p* suppression were further evaluated by histological assessments using articular cartilage explant cultures isolated from wild type mice. Both proteoglycan contents and type II collagen were markedly reduced by IL-1β, as demonstrated by an overall decrease in Alcian Blue/Hematoxylin (ABH) and Col2a1 immunohistochemistry (IHC) staining of cartilage explant sections. In addition, increased Mmp13 staining was observed in the presence of IL-1β (Figure 3c). However, knockdown of *miR-146a-5p* with its inhibitor largely reversed the imbalance between anabolism and catabolism caused by proinflammatory cytokine IL-1β, leading to a relatively normal cartilage matrix constitution, as further indicated by quantitative measures of ABH intensity (Figure 3c). Notably, NF-κB activation, exemplified by phospho-p65 staining, appeared to be dampened in the presence of the *miR-146a-5p* inhibitor (Figure 3d). Thus, these histological changes in matrix composition reflected the altered gene expressions observed in vitro. It is intriguing to speculate that increased *miR-146a-5p* correlates with an imbalance of anabolic–catabolic responses in articular chondrocytes as a result of NF-κB activation, and suppression of *miR-146a-5p* would protect against OA by modulating inflammation and maintaining the cartilage homeostasis.

### 3.4. Deletion of miR-146a in Chondrocytes Protects against Injury-Induced OA in Mice

Previous findings have demonstrated articular cartilage degeneration of *miR-146a* global knockout mice was ameliorated in spontaneous and injury-induced OA models [22]. It is thus tempting to speculate that a lack of *miR-146a-5p* in articular chondrocytes would alleviate OA development and lead to less severe OA. Recently, a *miR-146a* floxed mouse was generated [26], which provided a powerful tool to evaluate the tissue-specific role of this miRNA in regulation of a variety of tissue disorders and homeostasis. To this end, *Agc1Cre^ERT2^* mice were crossed with *miR-146a^f/f^* mice to generate *Agc1Cre^ERT2^;miR-146a^f/f^* (*miR-146a^Agc1^*, *miR-146a* LOF) mice. OA was introduced to littermate controls and *miR-146a* LOF mice at 3 months of age by meniscal ligament injury (MLI) surgery (Figure 4a). To avoid any development-related defects associated with ablation of *miR-146a* and to highlight the clinical relevance of the work, tamoxifen was administered to *miR-146a^Agc1^* mice and littermate controls 4 weeks post-MLI-surgery to induce *miR-146a* deletion specifically in chondrocytes.

As expected, by 10 weeks after injury, the control animals displayed severe OA phenotypes, including massive loss of cartilage, surface fibrillations and fissures, excess production of Mmp13, and diminished levels of Prg4 (Figure 4b,h). Additionally, apparent subchondral bone sclerosis was visualized in control mice by microCT assessment (Figure 4c,d). In contrast, OA in mice with silencing of *miR-146a* was substantially mitigated, as illustrated by the relatively smooth surface of articular cartilage, minimal staining of Mmp13, abundant expression of Prg4, and less thickening of the subchondral bone (Figure 4b,h). The delayed OA progression and ameliorated OA manifestations in *miR-146a* LOF mice were further evidenced by a significant decrease in Osteoarthritis Research Society International (OARSI) scores (Figure 4e) and subchondral bone volume over total volume (BV/TV) (Figure 4c,d). Consistently, with ex vivo findings shown by articular cartilage explants in response to IL-1β stimulation, silencing of *miR-146a* in vivo appeared to resolve, at least in part, NF-κB activation under injury conditions. It is important to note that uninjured knees with loss of *miR-146a* for 6 weeks did not display any phenotypic abnormalities (Appendix A), indicating that the upregulation of *miR-146a* under inflammatory conditions and further activation of NF-κB activity are partially responsible for development of more severe OA.

Since pain is one of the key debilitating symptoms in patients with OA, it is worthwhile to examine the pain response under the suppression of *miR-146a* as well. Two stimulus-evoked methods, including Von Frey and Small Animal Algometer (SMALGO) tests, were applied to evaluate pain behaviors associated with OA. Remarkably, compared to control animals displaying higher sensitivity to mechanical allodynia at both the plantar surface of the paw (Von Frey) and the injured knee joint itself (SMALGO), *miR-146a* LOF mice demonstrated less stimulus-evoked nociception and more tolerance and less pain response to these stimuli (Figure 4f,g). Together, these data suggest that articular cartilage with *miR-146a* LOF are more resistant to injury and less susceptible to the development of severe OA, shown by controlled NF-κB activity and maintenance of cartilage homeostasis.

### 3.5. Genes Potentially Targeted by miR-146a-5p in Articular Chondrocytes Are Enriched in Inflammatory Pathways and Cartilage Matrix Homeostasis

In order to explore the functional relevance of our selected *miR-146a-5p* in OA pathogenesis, we employed an integrated approach by combining a target prediction tool, TargetScan [31], and high-throughput RNA-Seq assays to search for potential gene targets and signaling pathways of *miR-146a-5p*. To gauge the response of chondrocyte transcriptomes to IL-1β, parallel RNA-Seq assays were performed in tandem with miR-Seq on IL-1β-treated primary murine articular chondrocytes. Expression profiling identified that 919 genes were significantly up- or downregulated in response to IL-1β (Log_2_FC > 2, *Padj* < 0.05) (Appendix A, Figure 5a), and pathway enrichment analyses with Enrichr [32] indicated their primary relevance to inflammation, cartilage matrix homeostasis, and chondrocyte metabolism (Figure 5b). Among the 420 downregulated genes, 55 were predicted targets of *miR-146a-5p* by TargetScan based on the confidence hit of the *miR-146a-5p* seed sequence in the 3′-untranlated region (UTR) of candidate genes (Figure 5c). Using the STRING database [33], we examined the functional association networks of gene subsets involved in the most significant pathways (a total of 189 genes) identified by enrichment analyses. Of the 55 putative gene targets of *miR-146a-5p*, *Cd28* and *Prkg1* were of particular interest, which showed direct or indirect interactions with a subset of genes highlighted in the most enriched pathways (Figure 5d,e).

Interestingly, when taking a closer inspection of the interaction networks, we found that *Cd28* and *Prkg1* were both linked to inflammatory pathways (Figure 5e). *Cd28* is implicated in T-cell activation, the induction of cell proliferation, and cytokine production [38]. Given the gene set of the most significant pathways as input, a functional association search using the STRING database presented a crosslink between Cd28 and several integrin family members of cell adhesion receptors, the latter of which play a major role in mediating chondrocytes–extracellular matrix (ECM) interactions in articular cartilage [39]. Indeed, expanded interaction networks primarily mapped pathways associated with ECM–receptor interactions and matrix remodeling (Figure 5e). In addition to ECM homeostasis, production of chemokines was stimulated, which presumably attracted inflammatory cells to the joint that further induced the secretion of inflammatory mediators and disease progression [40]. With regard to this part of the predicted association network, the *miR-146a*-*5p* targeted *Cd28* repression could be inferentially linked to cartilage matrix destruction, as indicated by overall upregulation of catabolic genes and downregulation of anabolic genes, as well as elevation of chemokines.

On the other hand, another predicted target of *miR-146a-5p*, cGMP-dependent protein kinase 1 (*Prkg1*), is a key mediator of the nitric oxide (NO)/cGMP signaling pathway [41]. Similarly, enriched protein–protein interactions with the same input revealed that *Prkg1* was functionally linked to mitogen-activated protein kinase 13, *Mapk13*, which was further mapped to a single major pathway, NF-κB signaling (Figure 5e). Notably, selected gene expression profiles, such as *Traf1*, *Tnfrsf(s)*, and *Nod(s)*, clearly indicate NF-κB signaling activation (Figure 5e). Nonetheless, some degrees of caution should be taken when interpreting these findings, since a bioinformatics approach was employed to predict the functional relevance of *miR-146a-5p* in OA pathogenesis.

## 4. Discussion

In this study, we firstly provided the evidence that inflammation, indicated by NF-κB activation, occurred in OA cartilage. In accordance with previous findings, we showed that the action of proinflammatory cytokine IL-1β, known for initiating a vicious cycle, led to elevation of other inflammatory cytokines, such as *Il6*, and excess expression of the catabolic enzyme *Mmp13* in articular chondrocytes. Given that miRNAs have emerged as key gene regulators in managing inflammatory processes in a variety of pathologies [42,43] and perturbations of miRNA levels are linked to onset and progression of OA [18], we next screened for miRNAs responsive to IL-1β by global miRNA expression profiling in primary murine articular chondrocytes. Of those miRNA extracted, *miR-146a-5p* was the most highly upregulated by IL-1β. In vitro and ex vivo data revealed that blocking *miR-146a-5p* greatly suppressed NF-κB activation and alleviated IL-1β-induced catabolic changes in cell cultures and cartilage explants. Consistently with these findings, our in vivo results further demonstrated that silencing *miR-146a-5p* specifically in chondrocytes, without causing phenotypic abnormalities, protected against injury-induced OA in mice. Finally, with parallel global mRNA and miRNA expression profiling, an integrated bioinformatics analysis implicated a strong association of *miR-146a-5p* with genes and pathways involved in inflammation and cartilage matrix homeostasis.

Our miRNA expression profiling from primary murine articular chondrocytes identified 23 differentially expressed miRNAs in response to IL-1β. Among these, *miR-146a-5p* was the most noticeably regulated (with nearly 4-fold upregulation) and was of the most interest, given a link between *miR-146a-5p* and susceptibility to a variety of inflammatory disorders. Previous studies have demonstrated the association between *miR-146a* and OA in rodents and human patients. In OA patients, *miR-146a* is identified as highly expressed in low-grade OA cartilage, coinciding with high levels of *MMP13* expression [36]. In vitro experiments further revealed that *miR-146a* is upregulated in response to IL-1β [44] or mechanical pressure injury [45] in murine and human chondrocytes. More importantly, mice with global knockout of *miR-146a* were resistant to spontaneous and surgical-induced OA pathologies, with less cartilage degeneration and more preserved cartilage homeostasis [22]. All of these observations suggest a beneficial effect of *miR-146a* inhibition in protection of OA progression. However, several groups also reported the contradictory effect of *miR-146a* in OA disease. *miR-146a* may serve as a novel therapeutic target for protecting cartilage from destruction in mechanical-injury-/inflammation-induced OA by promoting autophagy [46,47] or by negatively regulating inflammatory responses [48,49,50]. In this study, we found that abolishing *miR-146a-5p* significantly attenuated IL-1β-induced catabolic gene upregulation in primary murine articular chondrocytes and articular explants, and it also protected against articular cartilage degradation in injury-induced OA in mice. It is worthwhile to mention that (i) *miR-146a-5p* inhibition itself shows no side effect on articular chondrocytes in vitro and ex vivo, and (ii) the interference regimen of *miR-146a* in this study is of highly clinical relevance, as *miR-146a* was not modified prior to the onset of OA but rather at the early stage of the disease, highlighting the potential of targeting *miR-146a-5p* as a novel disease-modifying therapeutic to treat human OA.

*miR-146a* has been previously implicated to negatively regulate inflammatory responses by directly targeting *Traf6* and *Irak1* [37]. The present study, however, showed that *miR-146a-5p* inhibition antagonizes the activated proinflammatory pathways in articular chondrocytes. Here, combining a target prediction tool and gene subsets that are enriched in the most significant pathways from mRNA expression profiling, we searched for functional association networks using the STRING database and found that *Cd28* and *Prkg1* showed potential interaction with those highly regulated genes by IL-1β with high confidence. Notably, both of them pointed to the connection to inflammatory responses. *Cd28*, previously known as a paradigmatic T-cell co-stimulator, is identified not exclusively in T cells but also in other cell types. For instance, CD28-expressed macrophages exhibit a potent anti-inflammatory potential that limits their T-cell stimulatory capacity [51]. Prkg1, on the other hand, has recently been suggested to exert an anti-inflammatory reaction by reducing NF-κB activity and *Il1b* and *Nlrp3* gene transcription in the liver [52]. Hence, *miR-146a-5p* may interfere with the function of *Cd28* and *Prkg1* in the context of OA; however, this is speculative, since the direct binding of *miR-146a-5p* to these targets in chondrocytes remains to be experimentally determined. Moreover, targets of *miR-146a* have been studied in many different cell types, and it has become increasingly evident that regulatory effects of *miR-146a* are, to a certain degree, cell-type-specific and closely associated with physiological conditions.

With regard to the functional association analysis, one of the most interesting findings is that a significant portion of genes that were directly or indirectly associated with *miR-146a-5p* were mapped to inflammatory pathways, indicating *miR-146a-5p* plays a crucial role in mediating inflammatory responses in articular chondrocytes. Given that cytokines and inflammatory mediators associated with *miR-146a-5p* are markedly upregulated, it is reasonable to speculate that elevated *miR-146a-5p* expression in arthritic chondrocytes could be a factor that promotes an inflammatory OA state. Our results show that *miR-146a-5p* was increased in human OA cartilage and was robustly increased in response to IL-1β in articular chondrocytes. In the absence of *miR-146a-5p*, NF-κB response was dampened in vitro and in vivo, evidenced by the reduced *Il6* and phospho-p65, respectively. These findings argue that regulation of inflammatory response via silencing of *miR-146a-5p* is critical for proper resolution of NF-κB activity, particularly during the early hyperactive inflammatory phase of OA, whereas excess *miR-146a-5p* is poised to increase the amplitude and/or magnitude of NF-κB activation, leading to uncontrolled progress of inflammatory reactions and OA development. The findings that *miR-146a-5p* inhibition attenuated IL-1β-increased catabolism and alleviated injury-induced OA in mice strongly suggest *miR-146a-5p* is involved in inflammation-induced cartilage degeneration. This notion is further strengthened by the clearly skewed balance towards catabolism visualized in association networks in the case of upregulated *miR-146a-5p* (Figure 5). Nonetheless, as stated earlier, the principle aim of this study was not to identify miRNA-specific gene targets or establish mechanistic linkages but rather to elucidate the functional relevance of *miR-146a-5p* to OA-relevant phenotypes. 

In addition to findings about cartilage homeostasis, *miR-146a* is also linked to pain-related pathophysiology of OA [53,54]. Li et al. originally demonstrated *miR-146a* is downregulated in dorsal root ganglion and at the spinal dorsal horn, isolated from experimental OA rats that experience OA-evoked pain [53]. The reduced levels of *miR-146a* in these central compartments are closely associated with the upregulation of inflammatory pain mediators. Such linkage is further corroborated by their following study with gain- and loss-of-function of *miR-146a* in glia, the major cellular component of the central nervous system [54]. Our results show that loss of *miR-146a* in chondrocytes could relieve OA pain in an injury-induced OA mouse model. While the levels of *miR-146a* were not examined in glial cells, downregulation of some inflammatory pain mediators was observed upon removal of *miR-146a* from chondrocytes in vitro and in vivo. Thus, this miRNA regulates knee joint homeostasis and OA-induced algesia by simultaneously targeting inflammatory responses in cartilage and pain-associated factors in glial cells.

While our work highlights *miR-146a-5p* is a key downstream effector of inflammatory responses, especially the NF-κB pathway, in the regulation of articular cartilage homeostasis, the precise mechanisms of its actions and its specific molecular targets and biological functions remain to be determined. It is important to note our data do not exclude the possibility that additional downstream effectors of NF-κB signaling are involved in inflammation-induced disruption of articular cartilage homeostasis. Apart from *miR-146a-5p*, our miRNA expression profiling also identified 22 other differentially expressed miRNAs in response to IL-1β. Indeed, some of them had their roles in OA pathogenesis investigated, for instance, *miR-222* and *miR-210*. Members of the TNF receptor superfamily (TNFRSF), such as *Tnfrsf21*, are targets of *miR-210*, which is downregulated in OA [55,56], leading to aberrant activation of inflammatory pathways and induction of apoptosis. Additionally, *miR-222* decreases chondrocyte apoptosis and cartilage destruction by targeting histone deacetylase 4 [57], which is highly expressed in OA chondrocytes and potentially serves as a therapeutic target in OA [58]. It is thus speculated that the interplay of multiple miRNAs is crucial in the regulation of inflammatory responses of articular chondrocytes and the onset and progression of OA. Moreover, the ability to simultaneously fine-tune the expression of multiple genes makes miRNAs valuable targets and tools in complex diseases such as OA that involve interactomes of complicated molecular networks. 

## 5. Conclusions

Here, our transcriptome-wise miRNA expression profiling identified *miR-146-5p* as the miRNA most upregulated by IL-1β in murine articular chondrocytes. IL-1β-responsive *miR-146a-5p* is overexpressed in human OA cartilage. Knockdown of *miR-146a-5p* in chondrocytes antagonizes IL-1β-mediated inflammatory responses and increased catabolism in vitro and alleviates cartilage degeneration in an injury-induced OA in mice. Parallel gene expression profiling in tandem with miRNA-Seq from IL-1β-treated articular chondrocytes reveals that differentially expressed genes are enriched in pathways associated with inflammatory processes, cartilage matrix homeostasis, and cell metabolism. Prediction of *miR-146a-5p* gene targets and subsequent functional association analyses using a bioinformatics approach strengthen the functional role of *miR-146a-5p* in mediating inflammatory responses and regulation of cartilage matrix-related genes. In summary, our study employs a comprehensive approach combining genetic data with data from transcriptomics and provides compelling evidence that *miR-146a-5p* contributes to OA pathogenesis, possibly by acting as an accomplice of proinflammatory factors. The chondroprotective effects of silencing *miR-146a-5p* on the injury-induced OA in mice suggest that modulation of *miR-146a-5p* can be a potential therapeutic strategy to mitigate OA.

## Figures and Tables

**Figure 1 biomolecules-13-00123-f001:**
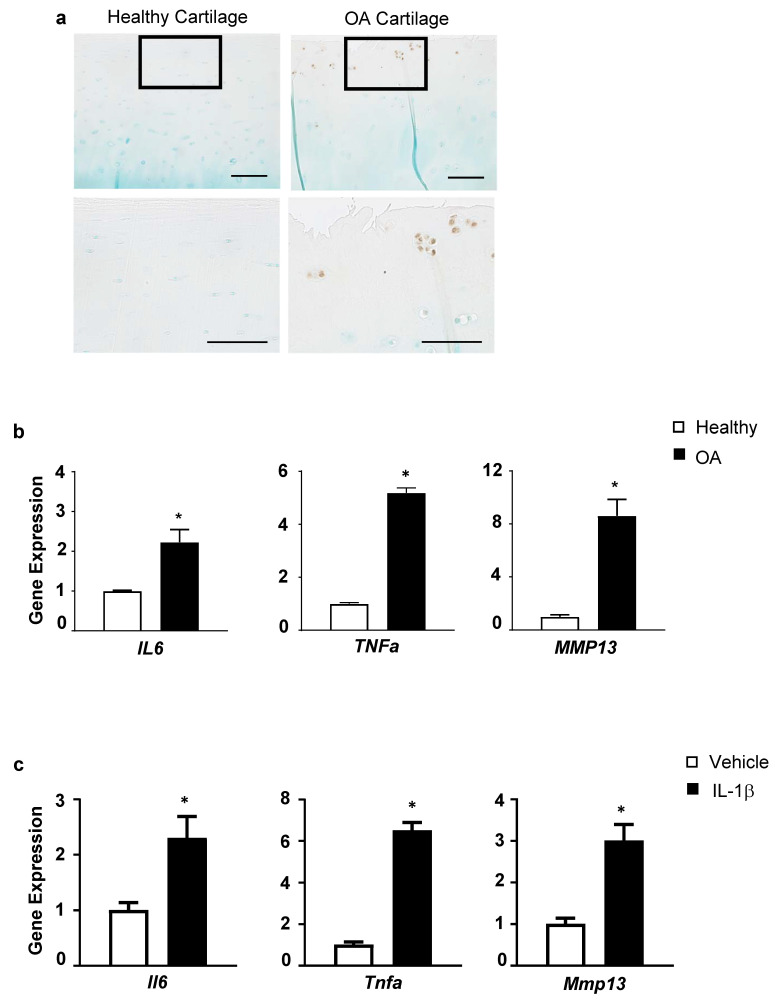
Inflammation occurs in OA cartilage, and IL-1β induces inflammatory responses and catabolic markers in articular chondrocytes. (**a**) Immunohistochemistry (IHC) for phospho-p65 on articular cartilage sections from healthy and OA patients (n = 3) (scale bar, 100 μm); (**b**) Real-time qPCR analyses of gene expression for *IL6*, *TNFα*, and *MMP13* in articular chondrocytes from healthy or OA patients (n = 3); (**c**) Relative gene expression for *Il6*, *Tnfα*, and *Mmp13* in primary murine articular chondrocytes treated with IL-1β or vehicle (n = 3). Data presented as mean ± SD. * *p* < 0.05 by Student’s *t*-test.

**Figure 2 biomolecules-13-00123-f002:**
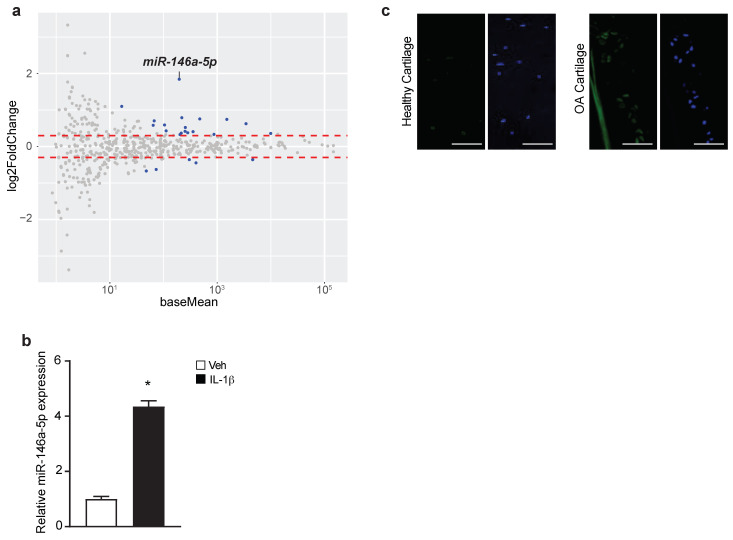
*miR-146a-5p* is induced by IL-1β in murine articular chondrocytes and is elevated in human OA cartilage. (**a**) MA plot of miRNA expression profiling from microRNA-Seq (miRNA-Seq) assays with IL-1β or vehicle-treated primary murine articular chondrocytes. Differentially expressed genes (DEGs) (Log_2_FoldChange > 0.3, *Padj* < 0.05) between IL-1β and vehicle groups were highlighted in blue (n = 4); (**b**) Real-time qPCR for *miR-146a-5p* in IL-1β- or vehicle-treated primary murine articular chondrocytes (n = 3); (**c**) Fluorescent in situ hybridization (FISH) for *miR-146a-5p* on articular cartilage sections from healthy donors or OA patients (n = 3) (scale bar, 100 μm). Data presented as mean ± SD. * *p* < 0.05 by Student’s *t*-test.

**Figure 3 biomolecules-13-00123-f003:**
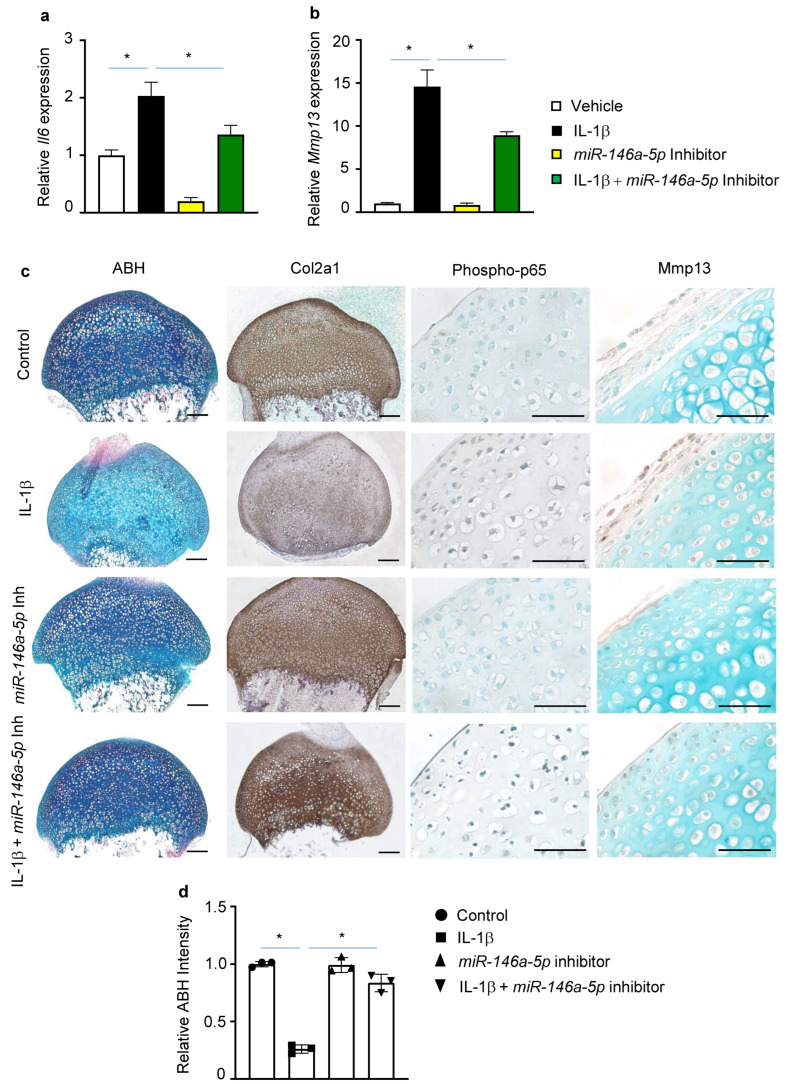
Suppression of *miRNA-146a-5p* antagonizes IL-1β-induced inflammation and catabolism articular chondrocytes. Relative gene expression analyses for (**a**) *Il6* and (**b**) *Mmp13* in IL-1β- or vehicle-treated primary murine articular chondrocytes in the presence of *miR-146a-5p* inhibitor or control inhibitor (n = 3). The data were normalized to vehicle-treated group; (**c**) Histological characterizations of IL-1β-treated murine articular cartilage explants in the presence of *miR-146a-5p* inhibitor, including alcian blue/hematoxylin (ABH) staining and IHC for Col2a1, phospho-p65, and Mmp13 (n = 3) (scale bar, 100 μm for ABH staining and Col2a1 IHC, and 100 μm for phospho-p65 and Mmp13 IHC); (**d**) Quantifications of relative ABH intensity on cartilage explant sections (n = 3). Data presented as mean ± SD. * *p* < 0.05 determined by two-way ANOVA followed by Tukey test to compare among multiple groups.

**Figure 4 biomolecules-13-00123-f004:**
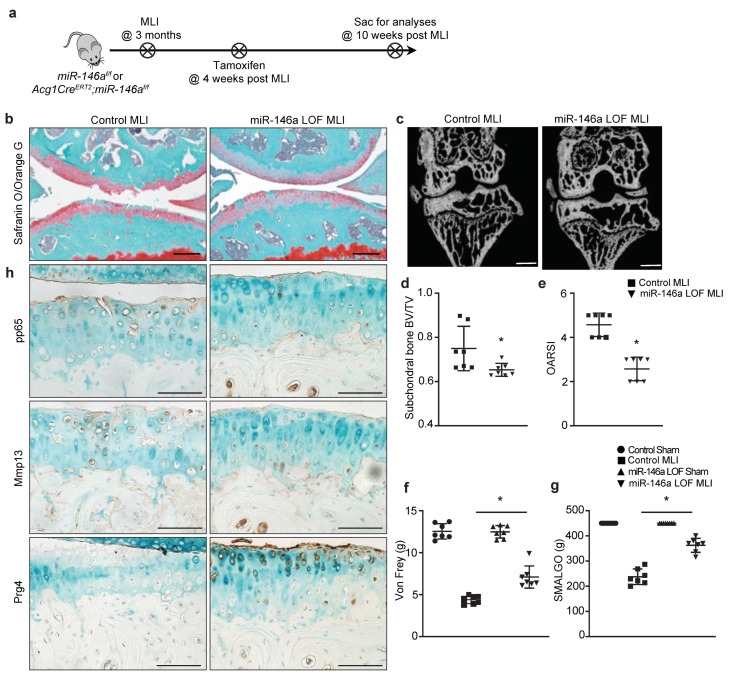
Loss of *miR-146a* in chondrocytes protects against injury-induced OA in mice. (**a**) Schematic illustration of in vivo experimental design; (**b**) Safranin O/fast green staining of knee sections of *miR-146a^f/f^* (control) and *Agc1Cre^ERT2^;miR-146a^f/f^* (*miR-146a^Agc1^*, *miR-146a* LOF) mice at 10 weeks following MLI surgery (n = 7) (scale bar, 100 μm); (**c**) MicroCT images of subchondral bone from control and *miR-146a* LOF mice at 10 weeks post-MLI-surgery (n = 7) (scale bar, 0.5 mm); (**d**) Quantifications of bone volume over total tissue volume (BV/TV) in reconstructed subchondral bone from control and *miR-146a* LOF mice at 10 weeks following MLI (n = 7); (**e**) Osteoarthritis Research Society International (OARSI) scores for the medial tibial plateau and femoral condyle from control and *miR-146a* LOF mice at 10 weeks following MLI (n = 7); OA-related pain assessed by (**f**) electronic Von Frey and (**g**) Small Animal Algometer (SMALGO) tests in control and *miR-146a* LOF mice at 10 weeks following MLI (n = 7). Data presented as mean ± SD. * *p* < 0.05 by Student’s *t*-test; (**h**) IHC for phospho-p65, Mmp13, and Prg4 on knee sections of control and *miR-146a* LOF mice at 10 weeks post-MLI (n = 7) (scale bar, 100 μm).

**Figure 5 biomolecules-13-00123-f005:**
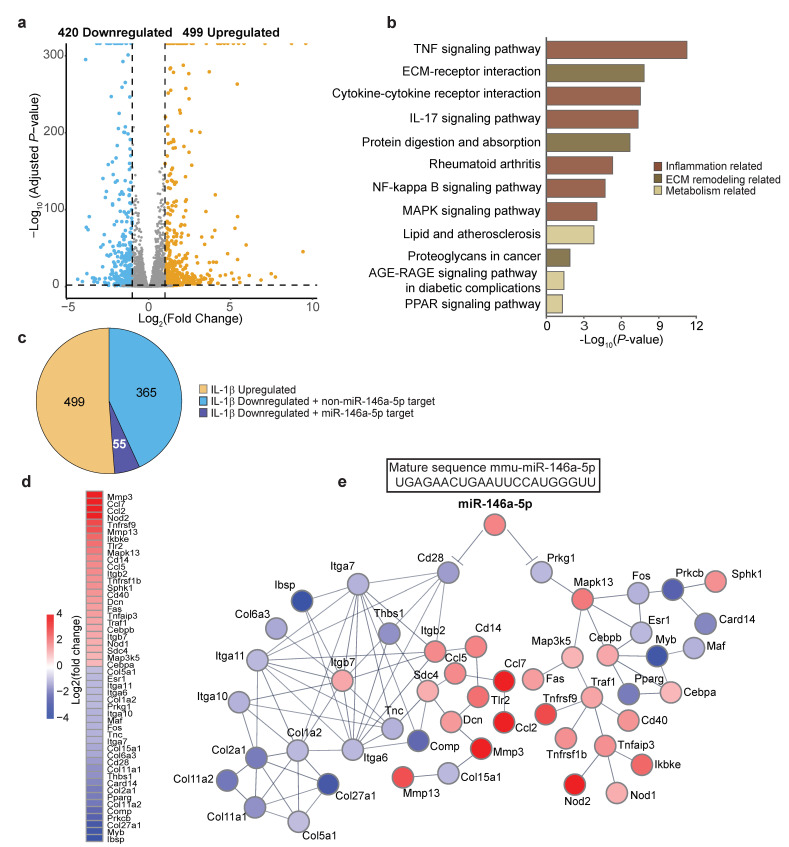
Putative gene targets of *miR-146a-5p* in articular chondrocytes are enriched in inflammatory pathways and cartilage matrix homeostasis. (**a**) Volcano plot of RNA-Seq showing the global transcriptional changes in IL-1β- versus vehicle-treated primary murine articular chondrocytes (n = 4); (**b**) Significant enriched KEGG pathways (*p* < 0.05) for differentially expressed genes (DEGs) (Log_2_FoldChange > 2, *Padj* < 0.05) between IL-1β- and vehicle-treated groups (n = 4); (**c**) Pie chart indicating the significantly IL-1β-regulated genes and putative gene targets predicted by the TargetScan web tool. Yellow represents IL-1β-upregulated genes. Light blue represents genes that are downregulated but not targeted by *miR-146a-5p*. Dark Blue represents genes that are downregulated and potentially targeted by *miR-146a-5p* in primary murine articular chondrocytes (n = 4); (**d**) Functional association network of *miR-146a-5p* and its gene targets linked to two major biological functions, inflammatory responses, and cartilage matrix homeostasis (n = 4); (**e**) Heatmap of DEGs illustrated in protein–protein interaction network (n = 4).

## Data Availability

The RNA-Seq and miRNA-Seq data generated in this study have been deposited in the Gene Expression Omnibus (GEO) database under accession GSE221485.

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
