# Peer review of "Silencing miR-146a-5p Protects against Injury-Induced Osteoarthritis in Mice"

_biomolecules, 2023, doi:10.3390/biom13010123_

Round 1
Reviewer 1 Report
Qin et al. discovered that miR-146a was upregulated in OA chondrocytes, which was also highly responsive to the treatment of IL-1b. Importantly, inhibiting miR-146a was shown to protect chondrocytes from IL-1 b treatment (in vitro) and ameliorate OA progression (in vivo). Lastly, bioinformatic analyses were performed to predict the targets of miR-146a. In general, this is an interesting study with conclusions supported by solid data. The manuscript was well-written. The reviewer only has several minor suggestions.
1. Figure 1a, please provide additional figures with higher magnification to better show the positively-stained areas.
2. Please keep consistent with the use of “TNFa” or “Tnfa”, “MMP13” or “Mmp13” in Figure 1b,c. BTW, the label “c” is missing in Figure 1.
3. Figures 2 b and c do not match how they were described in the figure captions. Also, the label of the Y-axis in the current Figure 2c has some issues. The sign that looks like a document should be “2”, right?
4. Line 271, it should be “healthy donors or OA patients.”
5. In the caption of Figure 3a, please state which group was used for normalization. It seems like the “Veh” group is the one.
6. There is no caption for Figure 3d. Please check and make corrections.
7. Figure 4, please make sure the figure captions correctly describe the corresponding figures.
Reviewer 2 Report
The manuscript by Qin et al. investigated the regulatory role of miR-146a in articular chondrocyte homeostasis associated with inflammation. The authors identified miR-146a as one of the most significantly upregulated microRNAs in normal primary murine articular chondrocytes in response to IL-1b. The authors employed a comprehensive approach combining genetic data with transcriptomics data and provided compelling evidence that miR-146a contributes to OA pathogenesis possibly by acting as an accomplice of proinflammatory factors. The chondroprotective effects of miR-146a silencing on the injury-induced OA in mice suggest that modulation of miR-146a can be a potential therapeutic strategy to mitigate OA.
In this manuscript, the authors provided sufficient evidence to demonstrate the critical role of miR-146a in regulation of articular chondrocyte homeostasis. Additionally, the in vivo data suggested the translational potential of this study. The findings in this study are novel and important. However, Guan et al. has recently reported that miR-146a is sufficient to protect OA pathogenesis by inhibiting Notch signaling in the joint. They propose miR-146a may be used to counter both aging-associated OA and mechanical injury-/inflammation-induced post-traumatic OA (Guan et al. Aging Cell, 2018: e12752). The discrepancy between this study and Guan et al.’s report needs to be explained.
Minor concerns:
1. The authors should clarify the accurate name of miR-146a. Is it “-5p” or “-3p”?
2. Line 30 “ regulating” should be “regulation.
3. Figure 1, the legend doesn’t match the figure itself. There is no panel “C” in the figure.
4. Figure 2, there are errors in the order of the panels, which is not consistent to the text. BTW, in the main text, the corresponding panel should be accurately labeled, instead of just using “Figure 2” for all the panels. This suggestion also applies to other figures.
5. Figure 3C, miR-146a inhibitor alone did not change the cartilage matrix constitution, but it improved the loss of cartilage matrix induced by IL-6. It may add some strength if the authors discuss this.
6. Line 463, “miR-146a may serve a novel therapeutic target”, the word “as” is missing after “serve”.
7. qPCR experiments are performed using human tissue or human cells?
8. The full gene list of RNA-seq and microRNA-seq needs to be provided.
Reviewer 3 Report
In this manuscript, Qin et al. reported a study to examine the molecular mechanism underlying the OA disease. OA is one of the most prevalent joint diseases, which causes significant disability in US and globally. Unfortunately, due to the unclear mechanism, currently there is no disease-modifying treatment for OA. Therefore, there is a critical need to better understand mechanism behind OA disease to develop potential therapeutic approaches.
In this manuscript, the authors have modeled articular chondrocytes with IL-1b treatment to mimic OA conditions, and then performed RNA-seq and microRNA-seq to comprehensively understand the epigenetic regulation in articular chondrocyte homeostasis under inflammatory conditions. miR-146a was identified as one of the most regulated microRNAs in vitro. The authors have further demonstrated the treatment efficacy by inhibition of miR-146a in cartilage explants and in mice. Altogether, Qin et al. comprehensively examined the epigenetic mechanism in articular chondrocyte lineage associated with joint inflammation, and demonstrated miR-146a as a potential target for OA treatment. The manuscript is well written with sufficiently in vitro and in vivo evidence. Several minor issues are listed below.
1. The raw data of RNA-seq and miRNA-seq needs to be available when requested or publicly.
2. The list of differentially expressed genes should be included in the manuscript.
3. Human articular chondrocyte isolation protocol needs to be included in the Methods.
4. The description of panel A in Figure 5 legend is missing.
5. Several typos need to be fixed.
